# Identification of earlier predictors of pregnancy complications through wearable technologies in a Brazilian multicentre cohort: Maternal Actigraphy Exploratory Study I (MAES-I) study protocol

Renato T Souza,[1] Jose Guilherme Cecatti,[1] Jussara Mayrink,[2]
Rafael Bessa Galvão,[1] Maria Laura Costa,[1] Francisco Feitosa,[3]
Edilberto Rocha Filho,[4] Debora F Leite,[4] Janete Vettorazzi,[5] Ricardo P Tedesco,[6]
Danielly S Santana,[1] Joao Paulo Souza,[7] on behalf of the MAES-I Study Group

For numbered affiliations see end of article.

**Correspondence to**
Professor Jose Guilherme Cecatti;
cecatti@unicamp.br

## ABSTRACT

**Introduction** Non-invasive tools capable of identifying predictors of maternal complications would be a step forward for improving maternal and perinatal health. There is an association between modification in physical activity (PA) and sleep–wake patterns and the occurrence of inflammatory, metabolic, pathological conditions related to chronic diseases. The actigraphy device is validated to estimate PA and sleep–wake patterns among pregnant women. In order to extend the window of opportunity to prevent, diagnose and treat specific maternal conditions, would it be possible to use actigraphy data to identify risk factors for the development of adverse maternal outcomes during pregnancy?

**Methods and analysis** A cohort will be held in five centres from the Brazilian Network for Studies on Reproductive and Perinatal Health. Maternal Actigraphy Exploratory Study I (MAES-I) will enrol 400 low-risk nulliparous women who will wear the actigraphy device on their wrists day and night (24 hours/day) uninterruptedly from 19 to 21 weeks until childbirth. Changes in PA and sleep–wake patterns will be analysed throughout pregnancy, considering ranges in gestational age in women with and without maternal complications such as pre-eclampsia, preterm birth (spontaneous or provider-initiated), gestational diabetes, maternal haemorrhage during pregnancy, in addition to perinatal outcomes. The plan is to design a predictive model using actigraphy data for screening pregnant women at risk of developing specific adverse maternal and perinatal outcomes.

**Ethics and dissemination** MAES-I has been reviewed and approved by each institutional review board and also by the National Council for Ethics in Research. Detailed information about the study is provided in the Brazilian Cohort website (www.medscinet.com/samba) and findings will be published in the scientific literature and institutional webpages.

### Strengths and limitations of this study

► This multicentre cohort will collect comprehensive data on major maternal and perinatal complications such as pre-eclampsia, small for gestational age/fetal growth restriction, preterm birth and gestational diabetes mellitus.

► Physical activity and sleep patterns will be estimated by an innovative wearable device used in the natural environment of the study subject.

► Physical activity and sleep patterns will be estimated from the beginning of the second half of pregnancy until delivery, covering a wide interval during pregnancy, allowing for the study of changes in physical activity and sleep patterns throughout pregnancy.

► One possible limitation is the first half of pregnancy at a time when this information was not covered.

## BACKGROUND

Reducing the global maternal mortality ratio to less than 70 per 100 000 live births by 2030 is one of the targets of the new United Nations Sustainable Development Goals.[1] Multiple challenges need to be tackled to achieve this target, but the 2016–2030 health and development agenda goes well beyond mortality reduction. The aim of the Global Strategy for Women's, Children's and Adolescent's Health is to ensure that every newborn, woman and child not only survives but thrives. This will only be possible if a transformative agenda centred on innovation is put into action.[2]

One of the major challenges lies in optimising earlier predictors and identifiers of maternal and perinatal complications. Delays

in diagnosing and managing maternal complications have been associated with poor outcomes.[3] Decreased self-perception of clinical signs related to maternal complications, difficulties in accessing the health system and poor quality of care may contribute to late identification of complications and a worse prognosis. The development of a non-invasive antenatal care (ANC) tool for identifying maternal subclinical signs during pregnancy may provide a window of opportunity for an earlier identification of abnormal patterns of physiological parameters related to pregnancy complications. Earlier identification occurs when recognition is made before clinical presentation by standard criteria based on clinical signs, symptoms and supplementary tests. Shortening the time between the onset of a complication and the initiation of appropriate management enables secondary prevention and reduction of maternal morbidity and mortality.[3–7]

Pervasive computing (ie, the trend towards embedding microprocessors in everyday-life objects so they can generate data) and wearable technology (ie, clothing and accessories incorporating computer and advanced electronic technologies such as sensor wristbands and/or waistbands) are ubiquitous and can generate a new dataset that requires correlation with pregnancy outcomes. Preterm birth and pre-eclampsia are two important pregnancy complications that have a relatively long subclinical phase before the appearance of signs or symptoms.[8 9] It is plausible that during subclinical phases of certain conditions the pattern of physical activity (PA) or sleep–wake rhythm is affected in some way and wearable devices could capture these changes. Although some studies have shown that PA patterns (actigraphy parameters) may be related to systemic inflammation and diseases in the general population,[10 11] there is a paucity of published literature that correlates wearable technology data with maternal complications.

The human circadian rhythm is regulated by endogenous physiological mechanisms and environmental stimuli.[12] Solid evidence indicates that modification in circadian rhythm or sleep and PA patterns are underlying conditions related to inflammatory, degenerative and/or metabolic chronic diseases such as diabetes, hypertension and cancer.[13] Circadian misalignment is defined as inappropriately timed sleep and wake, misplaced feeding periods and modification in PA behaviour.

Determining a cause or effect relationship between these modifications and the development of pathological conditions is a complex task. It seems that changes in appetite-stimulating hormones, glucose metabolism, inflammatory markers and mood are some of the related pathways.[13–15] Leproult et al[15] evaluated the effect of circadian misalignment on metabolic and inflammation markers in cardiovascular disease. Insulin action and release, and also levels of some inflammatory markers that are predictors of cardiovascular disease, were abnormal in individuals with circadian misalignment. The mechanisms involved in the association between changes in PA pattern and pathological conditions seem to have multiple aetiologies. Sani et al assessed circadian rhythms of more than 2300 African adult descendants. In addition to the evaluation of PA itself, the aim of those authors was to identify chronobiological patterns of adults from different socioeconomic settings. The study described that chronobiological behaviour can vary depending on individual BMI, socioeconomic background, work type and time of sunlight exposure. Many other factors, such as pathological conditions, may be potentially involved in a modification in chronobiological behaviour. Some metabolic, cognitive, cardiovascular and other chronic degenerative diseases have been associated with particular patterns of PA and sleep.[10 11 16–18] A previous observational study assessed various sleep parameters during pregnancy, that is, sleep onset latency (SOL), wake after sleep onset (WASO) and total nocturnal sleep time (TST). Difficulty in initiating sleep in early pregnancy was associated with higher body mass index, greater weight gain and higher blood pressure during pregnancy.[17] Palagini et al[19] reviewed the clinical evidence between chronic sleep loss and adverse pregnancy outcomes, discussing common mechanisms of stress system activation. Low-quality evidence suggests an association between sleep loss and prenatal depression, gestational diabetes, pre-eclampsia, abnormal length of labour, caesarean delivery, abnormal fetal growth and preterm birth. Those results corroborate with other findings regarding pregnancy and sleep disorders.[20–23]

Assessment of PA and sleep patterns can be performed by wearing small wrist (or waist) devices similar to a regular watch (actigraphy technology). More recently, substantial advance has been made in types of sensors, batteries, materials and output data, leading to lower cost, comfort, discretion and performance of the devices.[24] Nowadays, portable, lightweight devices have a large capacity to store data, including software with automatic scoring algorithm packages for the detection of wakefulness, sleep periods and PA.[24 25] Actigraphy estimation of PA and sleep patterns is validated as a proxy for chronobiological behaviour[26–29] and the use of an actigraphy device for 7–14 days provides reliable estimates of PA behaviour in older adults.[30–32] The performance of both hip and wrist devices has been shown to be reliable and acceptable for estimating PA and sleep–wake patterns.[33–36]

The main advantages of using wearable devices for actigraphy are non-invasiveness, 24/7 monitoring of PA and circadian patterns, and information about sleep habits and parameters in the natural environment of the subject.[24 25 28] We propose an innovative and strategic approach to monitor PA and sleep–wake patterns during pregnancy, establishing a large database comprised of clinical, epidemiological, PA and sleep–wake variables that are potentially capable of composing a prediction model for maternal complications during pregnancy. The main goal of this study is to identify earlier predictors of pregnancy complications by establishing a correlation between data on PA and sleep patterns using wearable devices (sensor wristbands) and maternal and perinatal complications and outcomes.

**Box 1  Participating centres in the Maternal Actigraphy Exploratory Study I**

► The Maternity of Centro de Atenção Integral à Saúde da Mulher (CAISM), University of Campinas, in Campinas, SP, Brazil;
  – Local principal investigator: Maria Laura Costa.
► Maternity of the Clinic Hospital, Federal University of RS, Porto Alegre, RS, Brazil;
  – Local principal investigator: Janete Vettorazzi.
► Maternity of the University Hospital, Jundiaí Medical School in Jundiaí, SP, Brazil;
  – Local principal investigator: Ricardo Porto Tedesco.
► Maternity of Clinic Hospital, Federal University of Pernambuco, in Recife, PE, Brazil;
  – Local principal investigator: Edilberto A Rocha Filho.
► Maternidade Escola Assis Chateaubriand (MEAC) – School Maternity of the Federal University of Ceará, in Fortaleza, CE, Brazil.
  – Local principal investigator: Francisco Edson de Lucena Feitosa.

## METHODS/DESIGN
### Study design
We will conduct a cohort study of 400 pregnant women using sensor wristbands capable of capturing information on daily PA and sleep patterns (exposure). This cohort study will be implemented in five ANC clinics linked to obstetric units in three different regions of Brazil that are already part of the Brazilian Network for Studies on Reproductive and Perinatal Health,[37] as shown in box 1. During an 8-month period, the ANC clinics will identify cases that are eligible to use the sensor wristband. Wearable technology data will be correlated with the occurrence of pregnancy and childbirth complications and outcomes, such as hypertensive disorders, gestational diabetes mellitus, fetal growth restriction and prematurity.

Eligible women will be identified up to 21 weeks of gestation and invited to participate in the study. A proper consent form will be applied and women who agree to participate will receive a sensor wristband to wear continuously from 19 to 21 weeks until childbirth.

### Study setting and population
Brazil is a multiethnic mixed-race population of diverse resourced settings.[38] Despite the high global overall human development index (HDI 0.727) in 2010, the HDI of Brazilian municipalities ranged from 0862 to 0418.[39] A mixed population is suitable for exploring information on patterns of maternal mobility and sleep, maximising external validity and comparisons to other populations. The following reasons support a study population of low-risk nulliparous women: (1) Previous obstetric history can refer to known risk factors for many maternal complications such as preterm birth, pre-eclampsia and diabetes.[13 40] Therefore, nulliparous women permit unbiased sampling regarding obstetric history. (2) Women with previous morbidities such as hypertension, diabetes, nephropathy or other chronic/degenerative diseases are more likely to present abnormalities in sleep–wake rhythms or PA patterns during pregnancy.

### Sampling
The five participating centres are regional referral obstetric units responsible for antenatal care of mainly high-risk pregnant women. Participating centres are listed in box 1. Nevertheless, there are primary health-care units strategically linked to these participating centres, enabling the identification and enrolment of women with non-pathological pregnancies. Recruitment strategies include approaching all eligible women in these participating centres and their linked facilities. An informed consent form will be applied for women who agree to participate.

### Eligible women: low-risk pregnant subjects
There is a lack of international consensus on criteria for low-risk pregnancies, although several factors are known to be associated with maternal and perinatal adverse outcomes. A recent study evaluating complications of 'low-risk' pregnancies of US Americans (10 million births from 2011 to 2013) indicated that 29% of low-risk women experienced an unexpected complication that required no routine obstetric/neonatal care.[41] This illustrates the difficulty in establishing a 'low-risk profile' for maternal/

**Box 2  Inclusion and exclusion criteria of Maternal Actigraphy Exploratory Study I**

**Inclusion criteria**
► Singleton pregnancy.
► Nulliparous (who had never given birth before).
► Between 19+0 and 21+0 weeks of gestation.

**Exclusion criteria**
► Unsure last menstrual period and unwilling to date the ultrasound.
► ≥3 Miscarriages.
► Major fetal anomaly/abnormal karyotype.*
► Essential hypertension treated before pregnancy.
► Moderate-severe hypertension at booking (≥160/100 mm Hg) or chronic hypertension using antihypertensive medication.
► Prepregnancy diabetes.
► Renal disease.
► Systemic lupus erythematosus.
► Antiphospholipid syndrome.
► Sickle cell disease.
► HIV or hepatitis B or hep C positive.
► Any condition that limits the performance of physical activity.
► Major uterine anomaly.
► Cervical suture.
► Knife cone biopsy.
► Ruptured membranes.
► Use of long-term steroids.
► Use of low-dose aspirin.
► Use of calcium (>1 g/24 hours).
► Use of eicosapentaenoic acid (fish oil) >2.7 g.
► Use of vitamin C ≥1000 mg and vitamin E ≥400 UI.
► Use of heparin/LMW heparin.
► Untreated thyroid disease.
► Use of antidepressant and/or anxiolytic agents.

*All information on fetal anomalies will be properly recorded.

perinatal complications. To make a better identification of eligible low-risk pregnant women, we excluded known potential confounders of prepregnancy conditions that could be related to adverse maternal or perinatal outcomes as shown in box 2, so we could assess PA and sleep patterns of a mostly 'normal' population. Nonetheless, features such as lifestyle habits and body composition (body mass index, height), and some non-severe chronic diseases including non-severe anaemia and/or asthma are not exclusion criteria in this study. However, these features and conditions may be a part of subgroup analyses (eg, composition of any previous disorder). Intra-individual and interindividual analyses of PA and sleep patterns can avoid possible bias by identifying potential confounders that may affect primary outcomes. A comparative analysis will be conducted, in which parameters of PA and sleep patterns will be collected in different stages of pregnancy from the same participant (intraindividual analysis) and compared with data collected at the same stage of pregnancy from different participants (interindividual analysis).

Eligible women are to be enrolled at 04–21 weeks of gestation. Inclusion and exclusion criteria are shown in box 2.

### Data collection methods

Essentially, Maternal Actigraphy Exploratory Study I (MAES-I) is composed of 4 key set points—three clinical visits during pregnancy and a postnatal visit. Clinical visits will be held at (1) 19–21 weeks, (2) 27–29 weeks and (3) 37–39 weeks. On the first, second, third and postnatal visits, additional information on maternal history, details of pregnancy complications, maternal biophysical data (weight, height, skinfolds) and adverse pregnancy outcomes will be collected following a specific standard operating procedure specially developed for MAES-I. Furthermore, the Perceived Stress Scale[42] and Resilience

Scale[43] will be applied during the 27–29 weeks visit. Figure 1 shows the set points of MAES-I.

Eligible women will be invited to use a 43x40x13 mm water-resistant wrist device similar to a regular watch (GENEActiv Original – Activinsights). The device contains an accelerometer for PA calculation and sensors for estimation of sleep–wake patterns by light and temperature measurements, using a proper software algorithm.

At the first set point of MAES-I (between 19+0 and 21+0 weeks of gestation), eligible women who agreed to participate will be instructed to wear the wrist bracelet device on the non-dominant wrist night and day (24 hours/day), uninterruptedly until childbirth (including bathing or recreational water activities). Participants will not need to press any buttons and functioning of the device requires no special care. The device will be configured to register PA and sleep–wake data automatically from the moment it is delivered to the participant during antenatal care visit. In addition, the battery charge will be held by the research assistant before delivering the device to the study participant.

The acquisition of actigraphy data can be performed in different frequencies (from 10 to 100 Hz). Since the frequency of data acquisition has an impact on battery life of the device (inverse relationship), measurement frequency will be set according to gestational age of the participant (table 1). This information will be registered in the database accordingly. Cumulative data will be downloaded during antenatal care visits, according to maximum return periods shown in table 1. Calculation of maximum return periods will be based on expected battery life. At each antenatal care visit, the used device will be returned to the research team and a new charged device will be provided to the participant.

A leaflet with detailed information and frequently asked questions about the device will also be provided. Women will also have a cell phone number to call in case of any

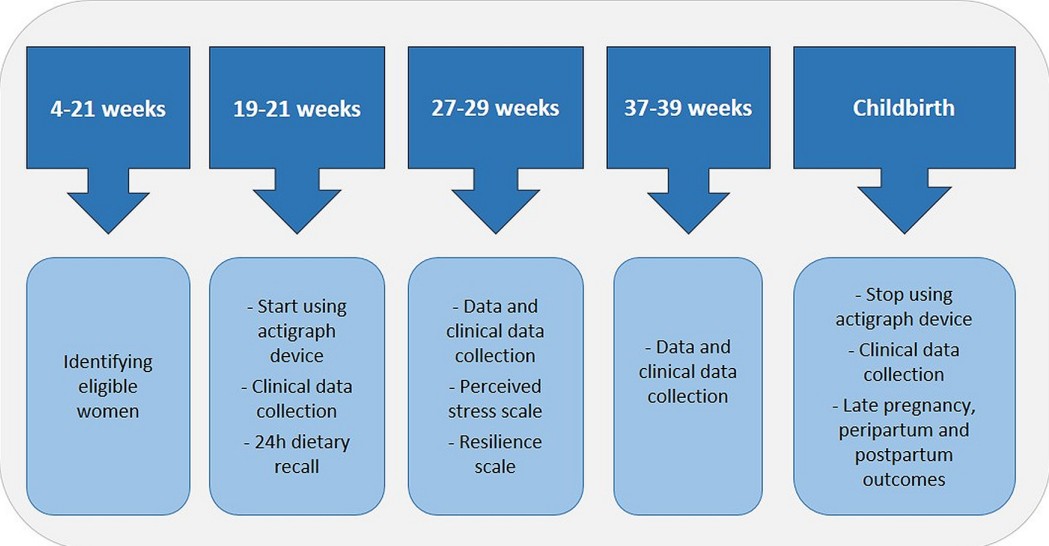

**Figure 1**  Set points of Maternal Actigraphy Exploratory Study I.

Table 1 Measurement frequency and maximum return periods according to gestational age—Maternal Actigraphy Exploratory Study I

| Gestational age (weeks) | Measurement frequency (Hz) | Maximum return period (weeks) |
|---|---|---|
| 19–32 | 20 | 4 |
| 33–36 | 30 | 2 |
| 37–42 | 50 | 1 |

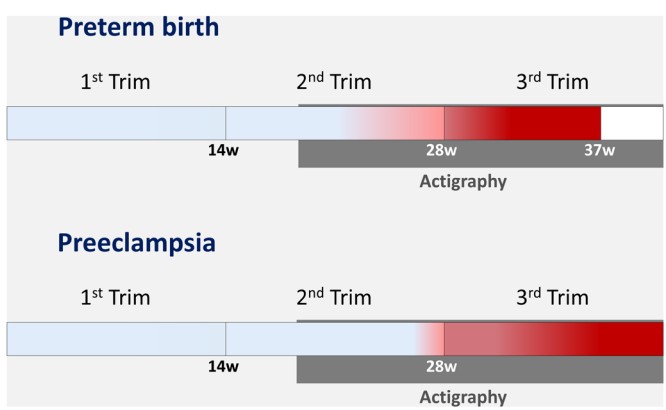

Figure 2 Estimated prevalence of preterm birth and pre-eclampsia according to gestational age (red represents the majority of cases) and evaluation period of physical activity and sleep patterns (in grey).

doubts regarding use of the device, or if any technical or medical concern arises.

During each antenatal care visit, the wrist device will be connected to a charge base which can be connected to a computer through an Universal Series Bus (USB) connection. All actigraphy data will be extracted to the computer as raw data '.bin' file. A proper open source software (Geneactiv Software) will allow the conversion of this file into '.csv' compressed epoch files for each 30 min of registered data, which can be read in Excel program. The actigraphy data will then be uploaded to an online database platform developed by MedSciNet, where all clinical study data will also be registered.

The actigraphy software uses several algorithms to translate numerical information obtained from epoch files into PA and sleep–wake patterns, which will compose the independent variables of this study. This is a centralised, secure, internet-based database that allows several procedures for prospective and retrospective monitoring, hierarchical access (local user, general manager, etc). The database will be translated into Portuguese and English, facilitating data collection for Portuguese-speaking teams and international monitoring. A correspondent paper form will be available for data collection if necessary (eg, internet connection failure for instance).

### Decision to start monitoring PA and sleep patterns between 19 and 21 weeks

There are various underlying mechanisms involved in the development of maternal and perinatal adverse outcomes that will be assessed, such as preterm birth, pre-eclampsia, gestational diabetes, fetal growth restriction and small for gestational age. Each disease may have a different preclinical phase, depending on environmental and individual aspects. In this phase, there are no clinical signs or symptoms. So far, the study of adverse maternal and perinatal predictors has been focused on early pregnancy (first trimester, <14 weeks of gestation) to maximise the window of opportunity for the performance of preventive interventions. However, we hypothesised that modifications in PA and/or sleep pattern due to underlying changes in maternal biological function might not be evident at a very early stage in pregnancy before the beginning of the preclinical phase. Our hypothesis is that changes might occur shortly before the manifestation of symptoms.

Furthermore, we took into account that major maternal complications, including pre-eclampsia, fetal growth

restriction and preterm birth, occur more commonly in late pregnancy and established the period between 19 and 21 weeks as an appropriate time to start assessment of PA and sleep patterns. A recent cross-sectional study conducted in 20 referral centres in Brazil, including the five participating centres of this proposal, showed that the occurrence of preterm birth before 28 weeks comprised less than 1% of all births and less than 8% of all preterm births.[44] In addition, the early onset of pre-eclampsia (before 34 weeks of gestation) complicates less than 0.4% of all pregnancies, according to a large retrospective cohort of more than 450 000 deliveries in the USA.[45] Figure 2 outlines the predicted prevalence of preterm birth and pre-eclampsia in the second trimester. Clinical presentation, when classic symptoms and signs of a certain disease/complication occur, is highlighted by pregnancy in red. Our hypothesis is that alterations in PA and sleep patterns may occur closer to clinical presentation, still in the preclinical phase when there are no symptoms or signs.

Briefly, an exploratory study required an arbitrary decision about the interval for monitoring PA and sleep patterns. To that end, we considered that: (1) the main maternal/perinatal complications of interest occur in the second half of pregnancy, more precisely in late pregnancy (figure 2); (2) any potential change in PA or sleep patterns occurred hypothetically days or weeks before the onset of maternal or perinatal complications. Then, we focused on monitoring women during the second half of pregnancy.

Thus, starting assessment at 19–21 weeks seems to be quite reasonable, providing a wide interval to monitor and predict major maternal and perinatal adverse outcomes.

### Actigraphy device

The actigraphy device that will be used for monitoring PA and sleep–wake patterns is the GENEActiv Original (GENEActiv, Activinsights, Huntingdon, UK). The device has multiple sensors including a microelectromechanical (MEMS) accelerometer, temperature (linear active

**Table 2** Performance of wrist and hip actigraphy methods according to different activities

| | Hip | | | Wrist | | |
| --- | --- | --- | --- | --- | --- | --- |
| | Sens | Spec | BA | Sens | Spec | BA |
| Sitting | 0.894 | 0.923 | 0.908 | 0.883 | 0.870 | 0.876 |
| Vehicle | 0.870 | 0.987 | 0.929 | 0.823 | 0.964 | 0.893 |
| Walking/running | 0.687 | 0.981 | 0.834 | 0.574 | 0.983 | 0.779 |
| Standing | 0.797 | 0.929 | 0.851 | 0.687 | 0.904 | 0.795 |
| Average | 0.812 | 0.955 | 0.881 | 0.742 | 0.930 | 0.836 |

Adapted from Ellis K *et al*.[34]
BA, balanced accuracy; sens, sensitivity; spec, specificity.

thermistor) and light (silicon photodiode) sensors, providing crude raw data for a variety of applications.

### Wrist versus waist wear: advantages and performance

Wrist-worn actigraphy devices are more comfortable to use during wake and sleep periods and provide the highest wear time compared with waist-worn monitors.[33 46] A non-systematic review published in 2011 showed that actigraphy is a useful and reliable tool to assess sleep patterns and circadian rhythm disorders, although there are some limitations in the diagnosis of sleep disorders or measurement of sleep stages.[25] Actigraphy had a very good concordance with polysomnography for assessment of sleep parameters in healthy subjects (ie, sensitivity >90% in estimating total sleep time). A recent study evaluated the concordance of PA estimation by wrist device in free-living settings in forty overweight or obese women.[34] Those women used both wrist and hip devices, and a small camera that captured participant behaviour for 7 days, monitoring PA behaviour (gold-standard comparison). There was a difference in hip and wrist machine learning classifiers, resulting from different methods/algorithms used to measure PA.[34] The sensitivity and specificity of hip and wrist estimations according to Ellis *et al*[34] are shown in table 2.

Two years previously, the same author published a similar evaluation of 40 adult women and men, showing that hip and wrist accelerometers predicted types of PA with an average accuracy of 92.3% and 87.5% respectively.[47]

Staudenmayer *et al*[48] investigated 20 participants who also wore two devices (wrist and hip), and concluded that wrist actigraphy can estimate energy expenditure in an accurate and relatively precise manner. Another study evaluated PA patterns in women at the top 40% or bottom 40% of the distribution of daily PA who wore wrist devices in a free-living environment. There was agreement in classification between hip and wrist accelerometers in about 75% of those women.[49] Additionally, total activity (counts per day) was moderately correlated (Spearman's r=0.73) with wrist-worn and hip-worn devices.

To the best of our knowledge, there are no systematic reviews or other high-quality evidence-based recommendations that support a particular method. Although a wrist-worn actigraphy device is not the most traditional

method, it might be the best choice for assessment of prolonged periods of PA or sleep patterns, considering that it performs similarly to a waist-worn device. The current proposal has no intention of diagnosing pathological behaviours or diseases, but it plans to identify different patterns throughout pregnancy and in different subgroups of women. Evidence suggests that wrist-worn actigraphy devices can accurately and more comfortably estimate PA and sleep patterns, mainly during prolonged periods and in free-living environments. Therefore, the MAES-I group adopted a wrist-worn device.

### Main variables

Independent variables assessed as potential predictors of maternal complications will be related to the sleep–wake cycle and mobility as:

#### 'Sleep' variables

SOL: time elapsed between full wakefulness and sleep.
TST: the amount of actual sleep *time* in a sleep episode (excluding time awake).
WASO: defined as total amount of time awake after sleep.
Sleep efficiency: the ratio between TST and time in bed.

The actigraphy device collects many pieces of information related to body position and body movements to estimate the described sleep variables. The actigraphy software will then be used to analyse data and generate output variables.

#### 'Physical activity' variables

Actigraphy technology estimates PA through various parameters collected by the actigraphy device. Briefly, according to Freedson *et al*,[50] the triaxial sensors stressed by acceleration forces can estimate movement intensity. The acceleration signal is converted to a digital signal and summed over a user-specified time interval (epoch). At the end of each epoch the activity count is stored. Then, according to count per minute (CPM) cut points, PA intensity can be categorised. The software translates information into quantitative variables using appropriate algorithms as follows:

Sedentary (hours/day): the number of hours per day when the CPM ranges from 0 to 99.
Light activity (hours/day): the number of hours per day when the CPM ranges from 100 to 1951.
Moderate activity (minutes/day): the number of hours per day when the CPM ranges from 1952 to 5724.
Vigorous activity (minutes/day): the number of hours per day when the CPM ranges from 5725 to 9498.
Very vigorous activity (minutes/day): the number of hours per day when the CPM is 9499–∞.
Metabolic equivalent (MET) rates: METs are also commonly used to express the intensity of PA. One MET is the energy cost of resting quietly, often defined by oxygen uptake as 3.5 mL/kg/min. MET rate expresses the working metabolic rate of subjects in

comparison to their resting metabolic rate. Briefly, the triaxial piezoelectric sensors stressed by acceleration forces can estimate movement intensity, converted to oxygen consumption required to perform such a movement.

Step counts/day: estimated step counts per day (estimated by proper algorithms using accelerometer data.)

## Outcomes

Primary outcomes are late pregnancy complications such as:

► Pre-eclampsia: hypertension after 20 weeks of gestation, systolic blood pressure (BP) ≥140 mm Hg and/or diastolic BP ≥90 mm Hg (Korotkoff V) on at least two occasions 4 hours apart with: (1) proteinuria 300 mg/24 hours or spot urine protein: creatinine ratio 30 mg/mmol creatinine or urine dipstick protein ≥ (+) OR, in the absence of proteinuria, hypertension and (2) any multisystem complication that are: haematological abnormalities; thrombocytopenia (platelets <100×10$^9$/L); disseminated intravascular coagulation and haemolysis; liver disease: increased aspartate transaminase and/or alanine transaminase >45 IU/L and/or severe right upper quadrant or epigastric pain, liver rupture; neurological problems: eclampsia, imminent eclampsia (severe headache with hyper-reflexia and persistent visual disturbance), cerebral haemorrhage; acute renal insufficiency: new increase in serum creatinine to >100 mmol/L antepartum or >130 mmol/L postpartum; pulmonary oedema confirmed by chest X-ray.[51]

► Gestational diabetes: new diabetes developing in pregnancy according to the WHO recommendation[52] that defines gestational diabetes as:
  – Fasting plasma glucose ≥92 mg/dL or
  – One-hour plasma glucose tolerance test (75 g load) ≥180 mg/dL or
  – Two-hour plasma glucose tolerance test (75 g load) ≥153 mg/dL.

► Spontaneous preterm birth: spontaneous onset of preterm labour or premature rupture of membranes leading to preterm birth, childbirth before 37 weeks of gestation.

► Provider-initiated preterm birth: defined as childbirth occurring at less than 37 weeks, medically indicated due to maternal/fetal compromise or both.

► Maternal haemorrhage: classified as (1) antepartum haemorrhage defined as bleeding from the genital tract after 24 weeks of gestation; (2) primary postpartum haemorrhage defined as the loss of at least 500 mL blood from the genital tract within 24 hours of childbirth.

Secondary outcomes include childbirth variables and neonatal adverse outcomes such as fetal death, caesarean section, small for gestational age (defined as birth weight below percentile 10 for gestational age), Apgar score <7 at 5 min, neonatal severe morbidity (table 3) and neonatal mortality before discharge.

## Plans for analyses

### Sample size estimation

This is an exploratory and innovative study focused on a specific population (pregnant women) and therefore there are no previously published parameters available for sample size estimation. Considering that the rate of pregnancy-related complications is 3%–20% (including pre-eclampsia, fetal growth restriction, gestational diabetes, haemorrhage, preterm birth, etc), assuming a large population (above one million pregnant women), an acceptable margin of error of 4%, involvement of five clusters (participating centres) and a 95% level of confidence, the study would require 384 women. Therefore, we rounded up this estimation to 400 initially low-risk pregnant women for enrolment in the study. We estimated the incidence of some main maternal complications considering the following studies:

Pre-eclampsia: an international prospective cohort study with nulliparous women termed Screening Of Pregnancy Outcomes (SCOPE) used similar criteria for low-risk profile, with a 5% of incidence of pre-eclampsia.[53]

Preterm birth: a recent cross-sectional study conducted in 20 referral obstetric centres in Brazil, including the five participating centres, showed that preterm birth was prevalent in 12.3% of all births.[44]

Gestational diabetes: in the previously mentioned SCOPE international cohort, the prevalence of gesta-

| Table 3 | Severe neonatal morbidity definition according to term/preterm status | |
|---|---|
| **Preterm** | **Term** |
| Grade III and IV intraventricular haemorrhage. | Grade II or III hypoxic ischaemic encephalopathy. |
| Chronic lung disease (home O$_2$ therapy or O$_2$ therapy at 36 weeks' gestation. | Ventilation >24 hours. |
| Necrotising enterocolitis. | Neonatal intensive care admission >4 days. |
| Retinopathy of prematurity, stage 3 or 4. | Apgar score <4 at 5 min. |
| Sepsis (blood or Cerebral Spinal Fluid (CSF) culture proven). | Cord arterial PH <7.0 and/or base excess >−15. |
| Cystic periventricular leukomalacia. | Neonatal seizures. |

tional diabetes was 8.9% in screened low-risk nulliparous women, according to the National Institute for Health and Care Excellence guidelines.[54]

Fetal growth restriction/small for gestational age: the previously mentioned SCOPE international cohort had a prevalence of 10.7% of small for gestational age newborns, according to customised centiles of birth weight (<10%).[55]

## Details of statistical analysis

According to the studies above, the predicted incidence of complications seems reasonable and reproducible in our cohort. Therefore, sample size estimation may ensure a sufficient number of cases of maternal and perinatal complications for the current proposal.

The epoch files obtained from Geneactv Software by reading data on sleep variables and PA parameters will be translated into numerical results and then averaged in 7 day periods. Therefore, only one value will be employed in statistical analysis for each variable per week of use of the wrist-worn device.

First, we will identify PA and sleep–wake patterns of women who did not develop adverse maternal or perinatal outcomes. This will permit the recognition of normal PA and sleep–wake patterns in a low-risk population without complications during pregnancy. We will use the same population to analyse changes in PA and sleep–wake patterns throughout pregnancy, allowing for gestational age periods.

Subsequently, we will compare PA and sleep–wake patterns of women who developed specific adverse maternal or perinatal outcomes with those who did not have any complications. Differences between groups may be identified and used as potential markers for specific pregnancy complications.

Afterwards, we will analyse changes in PA and sleep–wake patterns of women who developed adverse maternal or perinatal outcomes throughout pregnancy, comparing patterns in an attempt to discover which changes occurred before the onset of symptoms that could be related to pregnancy complications. If possible, we will conduct a subgroup analysis including a subpopulation with a potentially higher risk for maternal complications (confounder variables), including obesity, smoking and so on.

Finally, we will develop a predictive model for screening pregnant women at risk of specific adverse maternal and perinatal outcomes using PA and sleep–wake data estimated by actigraphy technology.

Analysis will be performed using the actigraphy software that translates collected information into PA and sleep–wake parameters. In addition, SOL, WASO and TST as well as sleep efficiency will be compared between participants throughout pregnancy using the Friedman and Wilcoxon tests for paired samples. The analysis of variance and t-test will be used to compare sleep parameters between participants per week of gestational age for repeated measures. The same tests will be applied

to analyse quantitative data on the median number of hours per day that different types of PA (sedentary, light, moderate, vigorous and very vigorous) are performed, MET rates and estimate of steps/day through the entire gestational period examined, and the comparison between participants per week of gestational age. Also, we will address the sensitivity, specificity and likelihood ratio for altered PA and sleep patterns or for their changes throughout pregnancy.

## Discontinuation of participants

Criteria for discontinuation include:
  Withdrawal of consent.
  Irregular use of the actigraphy device for prolonged periods, less than 50% of the whole planned time. Information of improper use of the device will be recorded if women notify the MAES-I team. Otherwise, the low level of use of the device will be observed after data discharge during antenatal care visits.
  Loss to follow-up, preventing us from downloading actigraphy data.

Women who decide to withdraw from follow-up care will be called by telephone and asked to return the wrist device. The last visit will be scheduled to regain the wrist monitor and direct the woman to a proper antenatal care service to continue medical consultations.

## Data and sample quality

All entered data will be prospectively and retrospectively monitored by local research assistants and a global monitor. Internal consistency of variables will be constantly performed by database and error messages are automatically flagged. A local research assistant will be responsible for checking all forms and actigraphy data before locking forms, assuring the good quality of data (ie, double-checking entered data and checking for inconsistencies between variables). The local principal investigator (PI) will be in charge of signing the case which will then be incorporated into the final database. The University of Campinas will coordinate, implement and monitor the study in the five participating centres. A general manager and a global monitor are also part of the coordinating team. The local team of each participating centre is composed of a local PI and research assistants.

## ETHICS AND DISSEMINATION

MAES-I focuses on low-risk nulliparous Brazilian pregnant women. Although classified as low risk for maternal and perinatal complications, these women are not free from suffering complications. Furthermore, first and second delays, defined as a delay in deciding to seek care and delay in reaching a healthcare facility,[56] are not uncommon. A barrier is created between earlier recognition of symptoms and timely intervention for the successful treatment of potentially life-threatening conditions. We believe that women will feel encouraged, empowered and willing to participate in a study aimed at developing a potentially useful prenatal care tool to identify the risk for maternal

and perinatal morbidity and mortality. Following national ethical regulations, the participants will not receive any financial compensation.

Women who agree to participate in the study will not have any disadvantage or difficulties in prenatal care. On the contrary, they will receive a contact number to find clinical researchers at any time (24/7 service), maintaining a closer contact with researchers and care providers. The MAES-I team is committed to contact healthcare providers if any potential complication arises.

Participating women will not be held accountable for any loss, theft or damage to the wrist device. These women will only be required to wear the device as a regular wrist-watch and no self-damage is expected.

Participating women will not be able to identify any PA or sleep parameters at any stage of the study. Data can only be downloaded through proper licensed software of the device. The actigraphy devices provided to participating women have a unique code which will be recorded in the database along with the interval of use per woman. Actigraphy data will be labelled using participant ID, device number, gestational age when the device was initially used and the return date of each device. Codes, ID number and numbers will ensure confidentiality of all participating women. The identity of all women will be kept confidential.

All women enrolled in the MAES-I cohort will sign an informed consent form.

Ethical principles of the Brazilian National Heath Council (Resolution CNS 466/12) will be upheld at every stage of this study. Anonymity of the source of information will be guaranteed and the woman will receive care irrespective of her agreement to participate in the study. The study also complies with the Declaration of Helsinki amended in Hong Kong in 1989. Methodological and ethical aspects of MAES-I protocol were developed following the Strengthening the Reporting of Observational Studies in Epidemiology guidelines.[57]

### Patient and public involvement

Patients and the public were not involved in this study for the development of the research question and outcome measures. However, the choice of a wrist device was based on user preference as reported. Participants of the study will have access to information available at the open-access website.

Detailed information about the study is provided in the Brazilian Cohort website (www.medscinet.com/samba). Publications of the results of the study can be found in the scientific literature and Institutional webpages. We intend to disseminate our findings to a scientific peer-reviewed journal, general free access website, specialist conferences and our funding agencies.

### DISCUSSION

Actigraphy is an innovative, non-invasive, non-operator dependent, wearable technology, that is capable of measuring diverse variables related to mobility, PA, sleep–wake and circadian cycle patterns under real-life conditions. Actigraphy devices have a high sensitivity in detecting sleep–wake parameters and are currently highly recommended by the American Sleep Disorder Association for diagnosis and therapy response of circadian rhythm disorders.[27 28 58] Although some studies show that using the actigraphy device for 7–14 days provides reliable estimates of PA behaviour in older adults, it is not absolutely clear how many days are needed to estimate habitual PA by using the wrist/waist device during pregnancy. In general, it seems to depend mainly on the type of actigraphy device, wear location and target population.[30 33] Nevertheless, MAES-I will provide sufficient data to assess different patterns throughout pregnancy.

The use of wearable PA monitors has increased considerably, owing to interest in the relationship between the pathophysiology of diseases and patterns of PA and sleep. A recent study on the use of PA monitors in human physiology research unravels current and potential use of the actigraphy device. The device can be applied in strategies that promote a healthier behaviour or predict outcomes.[59] The authors conclude that PA monitors, as well as other new 21st century technologies, have already transformed physiology research, revolutionising how we assess patients and opening new areas of interest. In addition, the use of objective measures to evaluate habitual sleep duration and outcomes in pregnancy is critical, considering recent reports of little agreement between objective and subjective assessments of sleep time.[60]

Alterations in sleep patterns, including less deep sleep and more nocturnal awakenings can be observed in pregnancy as early as in 10–12 weeks' gestation.[61] Sleep disturbances during pregnancy have been associated with preterm delivery, gestational hypertensive disorders, glucose intolerance and increased risk of caesarean delivery.[19] Shortened nocturnal sleep time was also associated with hyperglycaemia.[62] Persistent sleep deprivation has been correlated with depressive symptoms and stress perception by pregnant women.[61] These studies explored a correlation between PA patterns and sleep disturbances that determine complications through a well-established relationship between cause and effect. However, this correlation could not always be adequately determined due to study design.[17]

In a distinct manner, the intent of our analysis is to discover whether a maternal complication can be identified before the manifestation of its clinical signs, by evaluating PA and/or sleep patterns modifications of pregnant women. Considering existing evidence, we speculate that patterns of PA and/or sleep change days or weeks before clinical presentation of the complication. In general, the signs and symptoms of some maternal outcomes are part of the gold-standard criteria for diagnosis (high blood pressure, proteinuria and/or oedema in pre-eclampsia; premature contractions and cervical ripening/dilation in preterm birth; abnormal placental blood flow and insufficient fetal growth in intrauterine growth restriction). We

acknowledge that there are potential confounders and limitations in predicting maternal and perinatal complications using PA and sleep patterns estimated by actigraphy devices. The population in our research is expected to have different subgroups of women with different risks and associated factors contributing to maternal complications, such as obesity, smoking habit and with age under 20 or over 40 years old, for instance. None of those factors was considered an exclusion criterion. If possible, we intend to conduct a subgroup analysis of the maternal subgroups, since they may have different PA and sleep patterns. Nonetheless, we decided to adopt a pragmatic approach and not exclude such a common factor from our sample.

The use of actigraphy device during prenatal visits has the potential to become a new tool for monitoring pregnant women. It may improve maternal healthcare and identify altered PA and/or sleep patterns. Changes can be objectively measured by actigraphy before the occurrence of signs and symptoms. The focus is on providing new technology to monitor the development of potential maternal complications. Other positive points in our study are the data collection period (from 19 weeks until delivery) and the low-risk profile of the cohort, enabling us to describe PA and sleep patterns in a low-risk pregnant population and make a better interpretation of actigraphy data among pregnant women. Current clinical and biological predictors of major maternal complications such as pre-eclampsia, preterm birth, maternal haemorrhage and gestational diabetes still lack effective sensitivity and specificity.

If our hypothesis is confirmed, this will be an important step for introducing non-invasive screening procedures into prenatal care to identify women at higher risk for those conditions. Women could receive specific advice on the prevention and earlier detection of the condition, take immediate action and seek professional healthcare to receive appropriate treatment. This would avoid delays, the most significant factors contributing to low-quality healthcare in underprivileged women, which increase the still substantial burden of maternal morbidity and mortality. If we succeed in identifying 'specific patterns of physical activity and sleep' that are predictors of pregnancy complications, further validation studies are recommended to assess the effectiveness of screening procedures in management of these conditions. In addition, MAES-I will permit further specific studies among a high-risk population and also help to identify the best gestational age for monitoring, targeting a specific gestational age interval.

**Author affiliations**
[1]Obstetrics and Gynecology, Universidade Estadual de Campinas, Campinas, Brazil
[2]Faculdade de Ciencias Medicas, Universidade Estadual de Campinas, Campinas, Brazil
[3]Maternidade Escola, Universidade Federal do Ceara, Fortaleza, Brazil
[4]Obstetrics and Gynecology, Universidade Federal de Pernambuco, Recife, Brazil
[5]Obstetrics and Gynecology, Universidade Federal do Rio Grande do Sul, Porto Alegre, Brazil
[6]Obstetrics and Gynecology, School of Medicine of Jundiai, Campinas, Brazil
[7]Social Medicine, Faculdade de Medicina de Ribeirao Preto, Universidade de Sao Paulo, Ribeirao Preto, Brazil

**Acknowledgements** The MAES - I study group also included: Carina B Luiz and Luiza C Brust, School of Medicine, Federal University of Rio Grande do Sul, Porto Alegre, Brazil; Danilo Anacleto and Lívia C Nascimento, School of Medicine, Federal University of Pernambuco, Recife, Brazil; Daisy Lucena and Denise Ellen F Cordeiro, School of Medicine, Federal University of Ceará, Fortaleza, Brazil; Mariana B Rogerio, Departament of Obstetrics and Gynecology, Jundiaí Medical School, Jundiaí, Brazil.

**Collaborators** The MAES-I Study Group also includes: Carina B Luiz; Luiza C Brust; Danilo Anacleto; Lívia C Nascimento; Daisy Lucena; Denise E F Cordeiro; Mariana B Rogerio.

**Contributors** All authors contributed to the overall study design and specific methodologies. RTS, JGC, JM, MLC and JPS conceived the study design and wrote the first version of the study protocol. In a first meeting, the protocol was discussed and incorporated suggestions from RBG, FF, ERF, DFL, JV, RPT and DSS. RTS, JGC, JM and RBG planned the implementation of the study and developed the necessary material. RTS, JM, MLC and JGC drafted the manuscript that was afterwards revised by all other authors who gave suggestions. All authors discussed and made important contributions to the manuscript, read and approved the final version for submission.

**Funding** This study was funded by The Bill and Melinda Gates Foundation through the Grand Challenge Exploration program, call 19 (research grant OPP1182749).

**Competing interests** None declared.

**Patient consent for publication** Obtained.

**Ethics approval** MAES-I study has been reviewed and approved by the National Committee for Ethics in Research of Brazil (CONEP) and by the Institutional Review Board (IRB) of the coordinating centre (Letter of approval 1.834.116 issued on 24thth November 2016) and of all other Brazilian participating centres.

**Provenance and peer review** Not commissioned; externally peer reviewed.

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
