## [Reviewer comments · BMJ Open]

ARTICLE DETAILS

TITLE (PROVISIONAL)	The identification of earlier predictors of pregnancy complications through wearable technologies in a Brazilian multicenter cohort: Maternal Actigraphy Exploratory Study I (MAES-I) study protocol
AUTHORS	Souza, Renato; Cecatti, Jose; Mayrink, Jussara; Galvão, Rafael; Costa, Maria Laura; Feitosa, Francisco; Rocha Filho, Edilberto; Leite, Debora; Vettorazzi, Janete; Tedesco, Ricardo; Santana, Danielly; Souza, Joao Paulo

VERSION 1 - REVIEW

REVIEWER	Saori Morino Department of Physical Therapy, Faculty of Comprehensive Rehabilitation, Osaka Prefecture University, Japan
REVIEW RETURNED	06-Jun-2018

GENERAL COMMENTS	The study topic is interesting and important for allowing comfortable pregnancy by using a simple functional device. Below are comments and suggests for the authors to consider in their revisions. Some sentences were redundant and difficult to understand. e.g.) Additionally, to establish the period between 19-21 weeks as appropriate to start the assessment of PA and sleep patterns taking into consideration that the prevalence of the main maternal complications, as preeclampsia, fetal growth restriction, and preterm birth, are more common in late pregnancy. Lines 2, Page 5: Reference 10 is the research that investigated the relationship between sedentary time and some disease. Is sedentary time same as circadian rhythm? Lines 19-21, Page 8: Please clarify what is meant by “Intra and inter-individual analyses of PA and sleep patterns enable the identification of potential confounders affecting primary outcomes, avoiding potential biases.”? Lines 1, Page 9: Is there any reference about Standard Operating Procedure? Line 7, Page 9: To provide better clarity, change “19+0 - 21+0“ to “19-21” Lines 17, Page 9: Does the battery be changed to new one or charged during participant’s antenatal care visits? If it is charged,
---

pregnant women might charge it herself just in case. However, do you have enough time to charge it during their visits?

Lines 8, Page 10: The information about first trimester (~13 weeks of gestation) is needed for readers who are not familiar with this topic.

Decision to start monitoring: Decision of experimental period is important for this study, though it might be difficult. The reason of decision about the monitoring period is insufficient.

Lines 8-11, Page 10: Do you think modification of PA and/or sleep pattern might be risk factors for maternal complications? Why did you hypothesize that it possibly occurs shortly before symptoms?

Lines 25, Page 10: What is the "clinical presentation"? Does clinical phase start in 19-21 weeks of gestation? When is the clinical phase?

Lines 12-20, Page 11: What do you mean by these sentences? Does Hip better than wrist for monitoring of PA?

Lines 30-2, Page 11-12: What do you mean by "Although wrist wear of actigraphy is more conventional and accurate, it might not be the best choice for assessing long periods of PA or sleep patterns, even more considering the similar performance of the wrist wear."?

Lines 12, Page 12: Please change "Sleep onset latency (SOT)" to "Sleep onset latency (SOL)"

Lines 18, Page 13: How do you estimate step counts? Do you use acceleration data?

Lines 26, Page 13: Please change "disseminated" to "Disseminated"

Lines 25, Page 15: How did you decide the sample size as 384 women? Please give a calculation process

Lines 17-19, Page 15: Please clarify what is meant by "According to these studies, the predicted incidence of these complications, the leading causes of maternal and perinatal adverse outcomes, seems adequate for the current proposal and sample size estimation, although the complications are not cumulative."?

Lines 1-2, Page 16: The PA and sleep-wake patterns before onset of maternal complications should be investigated for discovering factors that related to onset of the complications and developing a predictive model for screening. Why do you compare that of before and after the onset of the complications?

Lines 13, Page 16: Is 50% of all planned time enough for monitoring PA? Is there any reference?
In addition, weekday and weekend, day and night should be separately considered in the criteria.

Lines 23, Page 16: Please explain in details or show some examples about "good quality of data".

Abbreviations: MET = metabolic equivalent. METS (or METs) = metabolic equivalents

	Table 2: Is there a formal definition that no previous delivery \geq 20weeks is nulliparous? Figure 2: Where is the GREEN? Does "Trim" mean "trimester"? What does shades of RED color mean? The incidence rate?
--	---

REVIEWER	Kirsti Krohn Garnæs Department for circulation and medical imaging, The faculty of medicine, NTNU, Norway. Department of Womans Health, St. Olavs Hospital, Trondheim, Norway.
REVIEW RETURNED	28-Jun-2018

GENERAL COMMENTS	Dear Souza and colleagues! This protocol describes a study including a large cohort of pregnant women, where you aim to use an actigraph for early prediction of maternal and perinatal health. It is very important to find effective and low-cost strategies to prevent and monitor risk for maternal and neonatal complications, and I find this study very interesting. Few studies have investigated the association between sleep-patterns and pregnancy risks. The protocol describes the study quite clear, however, I miss some clarifications in some parts of the manuscript. # Identifying early predictors of maternal complications are very important, and there is absolutely a need for more studies in this area. Further, I think it is very interesting to look at the association between physical activity/sleep pattern and maternal complications. However, I ask for the authors to make caution in use of the terms; "prediktive tool for maternal complications" "early predictors" and "innovative". The tool may predict the risk of maternal complications, more than the complications directly. There are probably so many contributing factors to the different complications listed in the manuscript, and also so many confounding factors affecting sleep pattern and physical activity. I also questions somewhat the term "early predictor". "Early predictors" for maternal health are in focus in todays research regarding the womens health before pregnancy and when she enter the pregnancy, so using this term for data collected from gestational week 19 might be questionable. I also ask for caution regarding frequent use of the term "innovative". Actigraph are often used in studies monitoring physical activity and sleep patterns in different types of population. # In the introduction part I miss more information regarding what type of sleeping patterns physical activity patterns are associated with maternal complications? This information is important to be able to claim that the actigraph could predict the different types of complications. # The plan for recruitment seems a bit vage to me. Could you please explain this process more detailed? Please provide us with more clarity regarding inclusion and exclusion criteria in the study. Are you going to include healthy women regardless of BMI? This is potentially important to state and to discuss. High BMI will regardless of "being healthy" increase the risk of all the maternal health outcomes you have stated to assess, and will also be directly associated with physical activity and sleep patterns. So it
---

	might be hard to distinguish weather risk for maternal complications is related to BMI or sleep and physical activity pattern. # Could the authors please provide more details into the statistical part of the manuscript? Power? What will be considered as statistical significant. How the sample size was calculated are not quite clear. # Please describe more in details how you are going to ensure confidential identity of all the women. # In the discussion part of the manuscript I miss more reflections regarding the possibility this tool has to actually predict maternal complications taking into account all the other factors despite sleep and physical activity that affects maternal risks, and all factors that affect sleep and physical activity patterns. # I would also ask the authors to consider providing questionnaires to all participants regarding sleep and physical activity patterns before pregnancy.
--	---

VERSION 1 – AUTHOR RESPONSE

Reviewers' Comments to Author:

Reviewer: 1

Reviewer Name: Saori Morino

Institution and Country: Department of Physical Therapy, Faculty of Comprehensive Rehabilitation, Osaka Prefecture University, Japan

Competing Interests: None declared

The study topic is interesting and important for allowing comfortable pregnancy by using a simple functional device. Below are comments and suggests for the authors to consider in their revisions.

Some sentences were redundant and difficult to understand. e.g.) Additionally, to establish the period between 19-21 weeks as appropriate to start the assessment of PA and sleep patterns taking into consideration that the prevalence of the main maternal complications, as preeclampsia, fetal growth restriction, and preterm birth, are more common in late pregnancy. We rephrased the sentence to “Additionally, we took into account that the occurrence of the main maternal complications, as preeclampsia, fetal growth restriction, and preterm birth, are more common in late pregnancy to establish the period between 19-21 weeks as appropriate to start the assessment of PA and sleep patterns.” Hope it was more readable.

Lines 2, Page 5: Reference 10 is the research that investigated the relationship between sedentary time and some disease. Is sedentary time same as circadian rhythm? “Circadian rhythm” was removed as it indeed did not refer to the statement and to the citation accordingly.

Lines 19-21, Page 8: Please clarify what is meant by “Intra and inter-individual analyses of PA and sleep patterns enable the identification of potential confounders affecting primary outcomes, avoiding potential biases.”? We added a sentence to better explain it. “It means that comparison of PA and sleep pattern parameters collected in different stages of pregnancy from the same participant (intra individual analysis) and collected at the same stage of pregnancy from different participants (inter-individual analysis) will be carried out.”

Lines 1, Page 9: Is there any reference about Standard Operating Procedure? Our group developed 5 different standard operating procedures for this cohort, but we do not intend to provide these documents now. Then, we do not provide any reference of them, considering they will appear later in another manuscript planned to describe all methodological steps of this study.

Line 7, Page 9: To provide better clarity, change “19+0 - 21+0” to “19-21”. We agree that using 19-21 would provide better clarity and that is why we preferred this way of describing gestation age inclusion interval. We’ve used this format in all sections of the manuscript, except in “Data collection methods” of Methods section and Table 2, where we preferred using 19+0 – 21+0 in order to clarify that pregnant women with 21+1 days will not be included. We understand that this clarification is needed at least at one sentence of the methods in order to avoid misinterpretation of the gestational age inclusion interval.

Lines 17, Page 9: Does the battery be changed to new one or charged during participant’s antenatal care visits? If it is charged, pregnant women might charge it herself just in case. However, do you have enough time to charge it during their visits? The following sentence was added (page 9) to clarify that a new charged device will be delivered at each antenatal care visit. “At each antenatal care visit, the used device will be returned to the research team and a new charged device will be provided to the participant.”

Lines 8, Page 10: The information about first trimester (~13 weeks of gestation) is needed for readers who are not familiar with this topic. We clarified this using the following sentence. “(first trimester, <14 weeks of gestation)”.

Decision to start monitoring: Decision of experimental period is important for this study, though it might be difficult. The reason of decision about the monitoring period is insufficient. As an exploratory study, we indeed needed to make an arbitrary decision and we tried to clarify what “variables/factors” were taken into consideration when making the choice. We added a new paragraph (page 10) as follows: “In brief, as an exploratory study, we indeed needed to make an arbitrary decision regarding interval of monitoring PA and sleep patterns. For that, we had taken into consideration: 1) the main maternal/perinatal complications of interest occur in the second half of pregnancy, more precisely in late pregnancy (Figure 2); 2) we hypothesize that any potential change on PA or sleep patterns might occur days or weeks before the onset of maternal or perinatal complication. Then, we focused monitoring women during second half of pregnancy.”

Lines 8-11, Page 10: Do you think modification of PA and/or sleep pattern might be risk factors for maternal complications? Why did you hypothesize that it possibly occurs shortly before symptoms? Our hypothesis is based on the concept that altered PA or sleep pattern would be an early manifestation of maternal/perinatal complications and not risk factor for the disease (stated at lines 9-12, Page 10). However, we understand that it might also be a confounder with risk factor as sedentary behavior is potentially related with chronic diseases and consequently with maternal/perinatal complications as well. Nevertheless, our analytical approach is focused in several steps (page 16): firstly, we will identify the different patterns of PA and sleep parameters of the studied population and only after that, we will compare changes throughout pregnancy between women who had complications and those who had not.

Lines 25, Page 10: What is the “clinical presentation”? Does clinical phase start in 19-21 weeks of gestation? When is the clinical phase? We added the following sentences to clarify “clinical presentation” and preclinical phase:

“when classic symptoms and signs of a certain disease/complication are presented” (Line 23, Page 10).

“preclinical phase when there is no symptoms or signs.” (Line 26, Page 10).

Lines 12-20, Page 11: What do you mean by these sentences? Does Hip better than wrist for monitoring of PA? There are different ways of wearing actigraphy devices. We thought it would be sensible to briefly describes the advantages and performance of the two main placement/wear and inform the rationale why we chose wrist wear. As stated in lines 9-18 of page 12, we chose wrist wear because it has been validated (compared to waist wear – “gold-standard”) and it is more comfortable.

Lines 30-2, Page 11-12: What do you mean by “Although wrist wear of actigraphy is more conventional and accurate, it might not be the best choice for assessing long periods of PA or sleep patterns, even more considering the similar performance of the wrist wear.”? Waist was incorrectly switched with waist and it made us miss the point. We tried to turn it more readable and giving the right message.

Lines 12, Page 12: Please change “Sleep onset latency (SOT)” to “Sleep onset latency (SOL)” Typo corrected.

Lines 18, Page 13: How do you estimate step counts? Do you use acceleration data? Accelerometer data can be used to estimate steps count by proper algorithms. “Step counts/day: estimated steps count per day (estimated by proper algorithms using accelerometer data.)”

Lines 26, Page 13: Please change “disseminated” to “Disseminated”. Change done.

Lines 25, Page 15: How did you decide the sample size as 384 women? Please give a calculation process. At the best of our knowledge, there is no similar study assessing PA and sleep patterns from 20 weeks of gestation to childbirth. We could not expect differences of maternal PA or sleep patterns within this interval – then, we cannot estimate sample size. As an exploratory study, we only could estimate outcome`s incidence – Lines 9-28, page 15. Power of analysis will be calculated retrospectively and it will be useful for future validation studies. Although we will enroll about 4 hundred nulliparous pregnant women, it is indeed an exploratory study.

Lines 17-19, Page 15: Please clarify what is meant by “According to these studies, the predicted incidence of these complications, the leading causes of maternal and perinatal adverse outcomes, seems adequate for the current proposal and sample size estimation, although the complications are not cumulative.”? The long phrase and lack of connectives made the sentence confusing. We changed it to “According to these studies above, the predicted incidence of these complications seems reasonable and reproducible in our cohort. Then, sample size estimation might assure enough cases of maternal and perinatal complications for the current proposal.”

Lines 1-2, Page 16: The PA and sleep-wake patterns before onset of maternal complications should be investigated for discovering factors that related to onset of the complications and developing a predictive model for screening. Why do you compare that of before and after the onset of the complications? The sentence was miswritten – we changed to “After that, we will analyze changes in PA and sleep-wake patterns of women who developed adverse maternal or perinatal outcomes through pregnancy, comparing the patterns and trying to discover which changes and when before the onset it would be related to pregnancy complications.”

Lines 13, Page 16: Is 50% of all planned time enough for monitoring PA? Is there any reference? The proportion of time using the device as an exclusion criteria was arbitrary chosen. There is no literature using similar methods to support this decision.

In addition, weekday and weekend, day and night should be separately considered in the criteria. Physical activity and sleep patterns will be addressed 24/7. We believe that considering weekday, weekend or night/day will introduce bias to our analysis.

Lines 23, Page 16: Please explain in details or show some examples about “good quality of data”. We gave two simple examples of how research assistants can assure good quality of data. Double-

checking entered data and checking for inconsistencies between variables are the main procedures performed by a local research assistant before saving forms. It assure reliability of entered data and avoid typo and other mistakes.

Abbreviations: MET = metabolic equivalent. METS (or METs) = metabolic equivalents. Changed accordingly.

Table 2: Is there a formal definition that no previous delivery \geq 20weeks is nulliparous?

Nulliparous is a medical expression for women who had never given birth before. We tried to make it clearer by changing the terms.

Figure 2: Where is the GREEN? Does "Trim" mean "trimester"? What does shades of RED color mean? The incidence rate? Sorry, but we did not realize that the green bar turned grey after editorial procedures. We've changed it in figure legend.

Reviewer: 2

Reviewer Name: Kirsti Krohn Garnæs

Institution and Country: Department for circulation and medical imaging, The faculty of medicine, NTNU, Norway. Department of Woman's Health, St. Olavs Hospital, Trondheim, Norway.

Competing Interests: None declared

Dear Souza and colleagues! This protocol describes a study including a large cohort of pregnant women, where you aim to use an actigraph for early prediction of maternal and perinatal health. It is very important to find effective and low-cost strategies to prevent and monitor risk for maternal and neonatal complications, and I find this study very interesting. Few studies have investigated the association between sleep-patterns and pregnancy risks. The protocol describes the study quite clear, however, I miss some clarifications in some parts of the manuscript.

Identifying early predictors of maternal complications are very important, and there is absolutely a need for more studies in this area. Further, I think it is very interesting to look at the association between physical activity/sleep pattern and maternal complications. However, I ask for the authors to make caution in use of the terms; "prediktive tool for maternal complications" "early predictors" and "innovative". The tool may predict the risk of maternal complications, more than the complications directly. There are probably so many contributing factors to the different complications listed in the manuscript, and also so many confounding factors affecting sleep pattern and physical activity. I also questions somewhat the term "early predictor". "Early predictors" for maternal health are in focus in todays research regarding the womens health before pregnancy and when she enter the pregnancy, so using this term for data collected from gestational week 19 might be questionable. I also ask for caution regarding frequent use of the term "innovative". Actigraph are often used in studies monitoring physical activity and sleep patterns in different types of population.

We understand that we do can estimate risks for developing maternal complications based on PA and sleep patterns, but that is not what we aim to assess. Physical activity and sleep patterns could be considered associated factors to maternal and perinatal complications and also indirectly related to some maternal phenotypes - in theory, obese women are more likely to be sedentary than normal BMI women, for example. However, our study design and analysis approach will indeed allow us to predict maternal and perinatal complications based on PA and sleep patterns during pregnancy. We will address sensitivity, specificity and likelihood ratio for altered PA and sleep patterns or for their changes throughout pregnancy. This is part of building a predictive model, as described in lines 18-

20, page 16. We added a sentence (line 24-25, page 16) to make it clearer. Nevertheless, we clarify that the current exploratory study will not be able to validate findings (we will not be able to validate a hypothetical predictive model at this stage).

Regarding the use of “early predictor”, we agree that this expression is more frequently used for early pregnancy (first trimester, for instance) and not for the interval we aim to start monitoring PA and sleep patterns. Therefore, we changed “early” to “earlier” and added the following sentence to make it clearer (Line 18-20, page 4). “We consider earlier identification when the recognition could be made before clinical presentation, when standard criteria based on clinical signs, symptoms, and supplementary tests are presented.”

Regarding the use of “innovative”, we have some argument for standing using it. Although there are many studies using actigraphy technology to estimate PA and sleep patterns, at the best of our knowledge, there is no such cohort study using actigraphy devices in pregnant women from 20 weeks to childbirth and aiming to predict maternal and perinatal complications. Then, we consider this a novel approach applied in an unmet need for maternal and perinatal health (prediction of maternal and perinatal complications).

In the introduction part I miss more information regarding what type of sleeping patterns physical activity patterns are associated with maternal complications? This information is important to be able to claim that the actigraph could predict the different types of complications.

We kept general information in introduction section (relationship of PA and sleep parameter with chronic diseases) and described more information regarding PA and sleep patterns associated with maternal complications in the discussion section. We highlight that there is a lack of studies about this relationship (PA and sleep patterns and maternal complications). Although biological pathways and mechanisms involved in this association are of great relevance, the majority of the literature supporting this relationship is based on epidemiological studies (more specifically in self-reported and short-term estimation of PA and sleep patterns). The current study intends not only to pragmatically address the relationship between PA and sleep patterns and maternal and perinatal complication, but also to describe the patterns throughout pregnancy (in the different stages of the second half of pregnancy). Lack of knowledge about this field has been published in the literature.

The plan for recruitment seems a bit vague to me. Could you please explain this process more detailed? Please provide us with more clarity regarding inclusion and exclusion criteria in the study. Are you going to include healthy women regardless of BMI? This is potentially important to state and to discuss. High BMI will regardless of "being healthy" increase the risk of all the maternal health outcomes you have stated to assess, and will also be directly associated with physical activity and sleep patterns. So it might be hard to distinguish whether risk for maternal complications is related to BMI or sleep and physical activity pattern.

We agree that maternal BMI is a potential confounder and it might play a significant role in the risk for maternal and perinatal complications (as it also related with PA and sleep patterns). In addition, we tried to describe in the “Eligible women: Low-risk pregnant subjects” section (page 8) how difficult and tricky the definition of “low-risk pregnant women” could be. However, we did not consider BMI an exclusion parameter as it would also introduce a bias to our sampling procedures and great part of our nulliparous population would not be represented (it is estimated that about 30-40% of pregnant women are obese or overweight in capitals in Brazil). Nonetheless, we clarified that women with high BMI will be part of a subgroup analysis. Also, controlling and adjusting for BMI and some other variables (lifestyle habits, for instance) will be considered.

Could the authors please provide more details into the statistical part of the manuscript? Power? What will be considered as statistical significant. How the sample size was calculated are not quite clear. At the best of our knowledge, there is no similar study assessing PA and sleep patterns from 20

weeks of gestation to childbirth. We could not expect differences of maternal PA or sleep patterns within this interval – then, we cannot estimate sample size. As an exploratory study, we only could estimate outcome`s incidence – Lines 9-28, page 15. Power of analysis will be calculated retrospectively and it will be useful for future validation studies. Although we will enroll about 4 hundred nulliparous pregnant women, it is indeed an exploratory study.

Please describe more in details how you are going to ensure confidential identity of all the women. We added the following sentences to clarify how participant confidential identity will be ensured (line 9-14, page 18). “Actigraphy devices provided for participating women have a unique code which will be recorded in the database together with the interval of use for each women. Actigraphy data will be labelled using participant ID, device number, gestational age when starting using each device and return date of each device. The use of such codes, ID`s and numbers will ensure confidential identify for all participating women. The identity of all women will be kept confidential.”

In the discussion part of the manuscript I miss more reflections regarding the possibility this tool has to actually predict maternal complications taking into account all the other factors despite sleep and physical activity that affects maternal risks, and all factors that affect sleep and physical activity patterns.

We added the following sentences in the discussion section to cover this: “We acknowledge the fact that there are potential confounders and limitations in predicting maternal and perinatal complications using PA and sleep patterns estimated by actigraph devices. We expect that our studied population will have different subgroups of women with different risks and associated factors playing a role on maternal complication. It includes obesity, smoking, extremes of age, for instance. None of those factors was considered exclusion criteria and, if possible, we intend to assess subgroup analysis for those maternal subgroups at they might present different PA and sleep patterns. Nonetheless, we decided to perform a pragmatic approach, not excluding such common factor from our sample.”

I would also ask the authors to consider providing questionnaires to all participants regarding sleep and physical activity patterns before pregnancy. We agree that providing questionnaires re sleep and PA patterns would be suitable/reasonable for such study which mains objective is based on PA and sleep patterns. However, once actigraphy devices are already validate for estimating PA and sleep parameters, we chose not to apply any other instrument for that. In addition, we do not intend to compare questionnaires with actigraphy data.

VERSION 2 – REVIEW

REVIEWER	Saori Morino Department of Physical Therapy, Faculty of Comprehensive Rehabilitation, Osaka Prefecture University, Japan
REVIEW RETURNED	29-Aug-2018

GENERAL COMMENTS	The manuscript has been correctly revised.
--

REVIEWER	Kirsti Krohn Garnæs Women's clinic, St. Olavs Hospital, Trondheim university hospital, Norway
REVIEW RETURNED	04-Sep-2018

GENERAL COMMENTS	General comment The manuscript is clearly prepared compared to the first submission. The aim of the study, method and outcomes are fully described. I really look forward to see the results of this interesting and very comprehensive study. I have just some minor comments: Abstract # Please introduce a clear research question in the end of the introduction part of the abstract, and clear outcomes in the method part of the abstract. Background # Study setting and population: I do not clearly understand the sentences starting in line 22-28, with «A few reasons might support the study population focused in low-risk nulliparous women:» As I understand it, you are arguing for only including nulliparous women? I totally agree this decision. But by typing “A few reasons might support” I read it like you don’t really have many reasons for focusing on only nulliparous women. Could you please re-write these sentences, clarify and simplify? Discussion # Could the authors please provide the manuscript with more reflections in discussion part regarding the study method: The participants will wear the actigraph from inclusion to delivery, will there be some disadvantages for the participants doing this? What about adherence and risk of drop-out? In addition, you describe in the method part the definition of “low-risk” pregnancy is unclear, and affect your inclusion criteria.
---

VERSION 2 – AUTHOR RESPONSE

Reviewer: 1

The manuscript has been correctly revised.

Thank you.

Reviewer 2

We appreciate Prof. Kirsti Krohn Garnæs interest and constructive suggestions.

General comment

The manuscript is clearly prepared compared to the first submission. The aim of the study, method and outcomes are fully described. I really look forward to see the results of this interesting and very comprehensive study. I have just some minor comments: Thank you. We will address all below.

Abstract

Please introduce a clear research question in the end of the introduction part of the abstract, and clear outcomes in the method part of the abstract.

As advised, in the Asbtract we stated a clear research question and disposed it more explicitly at the end of introduction. We also specified the outcomes to be evaluated in the study.

Background

Study setting and population: I do not clearly understand the sentences starting in line 22-28, with «A few reasons might support the study population focused in low-risk nulliparous women: As I understand it, you are arguing for only including nulliparous women? I totally agree this decision. But by typing “A few reasons might support” I read it like you don’t really have many reasons for focusing on only nulliparous women. Could you please re-write these sentences, clarify and simplify?

As for the Background section, we meant to say that there are some reasons why the study should focus on a population of nulliparous pregnant women. In order to make it clearer, we modified the sentence “A few reasons might support” to “The following reasons support the study population being focused in low-risk nulliparous women”.

Discussion

Could the authors please provide the manuscript with more reflections in discussion part regarding the study method: The participants will wear the actigraph from inclusion to delivery, will there be some disadvantages for the participants doing this? What about adherence and risk of drop-out? In addition, you describe in the method part the definition of “low-risk” pregnancy is unclear, and affect your inclusion criteria.

Regarding the Discussion, we found the suggestions about explaining if there would be disadvantages for the participants to use the wrist devices and about adherence and risk of drop-out to be extremely valuable, however we understand that these topics should be better detailed on the Method section. Because of that, we contemplated these subjects in subsections “data collection method” (page 8), “discontinuation of participants” (page 17) and “Ethics and dissemination” (page 18).

Regarding the term “low-risk pregnancy”, we meant to say that the concept of a low-risk pregnancy is not clearly established but, for the population described in this study, we understand that the participating pregnant women should not have some previous conditions before their inclusion, as it would be difficult to identify the outcome conditions evaluated (hypertensive disorders or gestational diabetes, for example) if the participants already have any of these diagnoses and if they had been already in treatment. Nonetheless, we modified the text in order to make it more readable and clearer.

VERSION 3 - REVIEW

REVIEWER	Kirsti Krohn Garnæs Department of Public Health and Nursing, Faculty of Medicine and Health Sciences, Norwegian University of Science and Technology (NTNU), Trondheim, Norway. 2 Department of Obstetrics and Gynaecology, St. Olavs hospital, Trondheim University Hospital, Norway.
REVIEW RETURNED	18-Jan-2019

GENERAL COMMENTS	My previous comments have been taken into account, and I have no further comments to the manuscript. I find the manuscript significantly improved, more structured and now well written.
--